# Risk map development for soil-transmitted helminth infections in Argentina

Eliana M. Alvarez Di Fino[1,2], Jorge Rubio[1], Marcelo C. Abril[2], Ximena Porcasi[1], María V. Periago[2,3]*

1 Mario Gulich Institute for Higher Space Studies, National University of Cordoba, National Commission of Space Activities (UNC_CONAE), Córdoba, Argentina, 2 Fundación Mundo Sano, Buenos Aires, Argentina, 3 Consejo Nacional de Investigaciones Científica y Técnicas (CONICET), Buenos Aires, Argentina

* vperiago@mundosano.org

## Abstract

### Background

Soil-transmitted helminths (STHs) comprise a group of helminth parasites that are included in the list of Neglected Tropical Diseases and require a passage through the soil to become infective. Several studies have detected that infection with STHs are associated with certain socioeconomic, environmental and soil characteristics. In Argentina, the presence of these parasites has been detected through a few point studies conducted in localities from 11 of the 23 provinces that comprise the country.

### Methods/Principal findings

The most important characteristics previously associated with the presence of STHs were identified and ranked through the use of an expert survey and the Analytical Hierarchy Process (AHP) in order to construct a risk map of STHs specific for Argentina. Prevalence data from previous studies was used to validate the generated risk map. The map shows that half of Argentina, from the Central provinces to the North, contains localities with the characteristics necessary for the development of these parasites.

### Conclusions/Significance

The predicted map should serve as a useful tool for guiding the identification of survey areas for the generation of baseline data, detecting hotspots of infection, planning and prioritizing areas for control interventions, and eventually performing post-implementation surveillance activities.

**Data Availability Statement:** All relevant data are within the manuscript and its Supporting Information files.

## Author summary

Given the lack of information on the prevalence of soil-transmitted helminths (STH) in Argentina, the aim of this study was to generate a national risk map to guide the implementation of surveys to determine baseline data. For this purpose, factors most associated

**Funding:** This study was funded by Fundación Mundo Sano. Mundo Sano did not play any role in the study design, data collection and analysis, decision to publish or preparation of the manuscript.

**Competing interests:** The authors have declared that no competing interests exist.

with the presence of STHs from previous studies conducted in other countries were ranked by experts in the field through the use of an Analytical Hierarchy Process. These characteristics were grouped into categories: socioeconomic, environmental and soil characteristics. The resulting map was validated against point prevalence studies conducted in the country and it showed that the top half of the country, from the Central provinces to the North, have localities with conditions appropriate for the development of these parasites.

## Introduction

Neglected Tropical Diseases (NTDs) are endemic in tropical and subtropical areas of the world and include 20 diseases [1]. Among them are a group of parasites called soil-transmitted helminths (STHs) that are exclusively human (no animal reservoir) and which require a passage through the soil to become infective. The species that compose this group may be divided into those that infect through the fecal-oral route, *Ascaris lumbricoides*, *Trichuris trichiura*, and those that infect via penetration of the skin, the hookworms (*Ancylostoma duodenale* and *Necator americanus*) and *Strongyloides stercoralis*. This last species is not included in the list due to certain unique characteristics that make it hard to detect, quantify and treat [2] using World Health Organization (WHO) guidelines. For control of the morbidity caused by infection with these parasites, WHO guidelines suggest preventive chemotherapy (PC) with albendazole or mebendazole in high-risk groups: pre-school, school-aged children and women of childbearing age. Deworming should be performed once a year if the prevalence is between 20 and 50%, twice a year if it's more than 50% and it should be individualized in those communities with less than 20% prevalence [3].

Studies conducted in different parts of the world have detected that infection with STH is associated to certain characteristics that may be grouped basically into socioeconomic and environmental. For example, these parasites are more prevalent in populations that lack access to safe water, improved sanitation and hygiene (WASH), those that live in precarious houses with overcrowding, and those without waste management [4–12]. Some research groups have tried to create risk maps by identifying associations with environmental variables such as temperature, precipitation, humidity, NDVI, elevation, land cover in different countries or regions of the world [11,13–17]. Associations with soil characteristics such as composition, bulk density, organic carbon content, moisture and acidity have also been used in these maps [18–23].

Based on these past experiences, in the current study, the most important socioeconomic, environmental and soil factors previously associated with the presence of STH were identified. A survey was created in order to obtain the input from different experts in the field to rank them in order of importance from least to most important. The answers from the different participants of the survey were used to create a matrix using the Analytical Hierarchy Process (AHP) originally developed by Saaty [24] in order to construct a risk map for STHs specific for Argentina.

In Argentina, the presence of all five species of STHs have been detected through point studies conducted in a few cities spread throughout 11 of the 23 provinces of the country and the prevalences found in these studies vary from very low to very high depending on the province, the locality within the province and the coprological technique used for analysis of the fecal samples (more or less sensitive). The prevalence data obtained from these studies was used to validate the generated risk map and a machine model approach using a decision tree (DT) algorithm was used to determine the predicted prevalence for each city to see if it

matched with the prevalence obtained from the studies. Therefore, the aim of this study is to identify high risk areas in order to be able to prioritize surveillance of STHs and subsequently be able to plan effective control interventions in those places where the presence of these parasites is verified.

## Methods

### Study area

Argentina is located in the southern cone of South America (minimum and maximum longitude: -73.580 and -53.590; minimum and maximum latitude: -55.050 and -21.780), covering a total area of 3,761,274 km$^2$. In this study, we analyzed the risk for STH infection in 3526 cities of Argentina that correspond to points of census cities provided by the Argentinian Institute of Statistics and Censuses (INDEC); since not all cities count with the variables used to create the risk map.

### Identification of parameters

A literature review was performed in order to identify the most important factors associated to the prevalence of STH infections and the main ones are summarized in Table 1 [13,20–22,25,26]. These were grouped by categories into socioeconomic (SEC), environmental (EC) and soil characteristics (SC).

An AHP was used as a multiple criteria decision making tool to determine the importance of these factors for the development of STH infections. This methodology introduced by Saaty [24] allows breaking down a complex problem into components and arranging them in hierarchic levels [27]. A pairwise comparison judgment matrix that included the 19 factors identified through the literature review was used to survey experts in the field of STHs and members of the STH Coalition (http://www.childrenwithoutworms.org/sth-coalition). The participants were asked to evaluate the relationship between infection with STH and the factors from each of the categories from Table 1 rating them in importance from 1 to 9. The ranking system was as follows: 1 designated equal importance; 3 moderate importance of one vs the other, 5 essential or strong importance, 7 very strong importance, 9 extreme importance and even numbers were designated as intermediate values between two adjacent judgements [28]. From the eigenvalues of this matrix, the coefficient for each group of characteristics was obtained in order to generate three different maps.

The risk associated to SEC, EC and SC was determined by a linear combination of the factors mentioned in Table 1, using the following general equation:

$$Risk = Coef_{F_1} \times F_1 + Coef_{F_2} \times F_2 + \ldots + Coef_{F_N} \times F_N$$

**Table 1. List of the most important factors associated with soil-transmitted helminth infections grouped into categories.**

| Socioeconomic characteristics (SEC) | Environmental characteristics (EC) | Soil characteristics (SC) |
|---|---|---|
| • F1 = % Population with sanitation at home<br>• F2 = %Urban population<br>• F3 = % Population with access to safe drinking water<br>• F4 = % Population with computer<br>• F5 = % Population with overcrowding<br>• F6 = Unemployment rate | • F1 = Normalized Difference Vegetation Index (NDVI)<br>• F2 = Maximum temperature<br>• F3 = Minimum temperature<br>• F4 = Altitude<br>• F5 = Precipitation driest month<br>• F6 = Annual precipitation<br>• F7 = Land Cover type<br>• F8 = Precipitation wettest month<br>• F9 = Mean annual temperature | • F1 = Soil acidity<br>• F2 = Bulk density<br>• F3 = Organic carbon content<br>• F4 = Gypsum content |

Where Risk represents total risk associated to each characteristic, determined by the addition of each factor (F) multiplied by its coefficient (Coef). For SEC the risk was calculated using F1 to F6, as seen in Table 1, for EC using F1 to F9 and for SC using F1 to F4.

## Data collection

To characterize the socioeconomic situation of each city, data were obtained from the last national census performed in 2010 (data processed with Rdatam +Sp) [29,30]. The census was performed through a questionnaire and personal interview in all the localities of the country regardless of its population size [29]. This value represents the average of each city; therefore, in big cities, there may be areas of a locality that have values well below average and areas of a locality that have values well above average, and on the contrary small villages could be aggregated to the nearest city for social data. Therefore, not all census data are exactly georeferenced and the resulting map is not continuous.

Environmental and soil data were obtained from WorldClim version 2 [30] at 30 arc seconds (~1000 m) for a large time series (1970 to 2000). Coverage of data from WoldClim is available until -50˚S latitude and this is why the maps generated through this software appear to be cut at that point. In addition, the state of vegetation was included through MODIS TERRA Normalized Difference Vegetation Index (NDVI) products (MOD13 A2 version 5) distributed by NASA´s Land Processed Distributed Active Archive Center (LP DAAC) were obtained for a ten-year period (2004 to 2014) and the mean of the warmest month was calculated and used as a general vegetation approach (January NDVI mean). Land cover type was obtained from the Latin American SERENA project [31], containing 23 main categories of land cover (needle leaf and broadleaf forest, barren land, grassland, wetlands, crops and urban areas, among others). To evaluate the cover diversity around each city, a diversity approach was calculated using a 9x9 pixel window as the number of different covers observed within that window [32]. Finally, altitude was obtained from the Space Shuttle Radar Topography Mission (SRTM), available from the NASA-Earth data repository (https://search.earthdata.nasa.gov/), with a spatial resolution of 3 arc seconds (~90 m) and vertical overall accuracy of 12 m.

Data for the soil characteristics were obtained from a Soil Atlas [33] for Argentina from the National Institute of Agricultural Technology (INTA) at a scale of 1:500,000. This Atlas presents regions of the different soil orders, 12 in total [34], which have different characteristics of acidity, density, organic carbon content and gypsum. These characteristics are categorized as described in Table 2, therefore in Table 3 the characteristics for each order of soil used in this study is described. Only eight of the 12 orders of soil are present in Argentina, therefore the table only lists those that correspond.

## GIS analysis

QGis software was used to develop the risk maps for each characteristic [35]. Risk information is represented through vector layers for the three groups of characteristics in each city of Argentina. A summary of the methodology is presented in **Fig 1**.

**Table 2. Categorization of soil characteristics based on standard measurement and the measurements normalized and used in the current study.**

| Soil Characteristic | Standard measurement | Normalized for this study |
|---|---|---|
| Acidity | >7, 7, <7 | 1, 0, -1 |
| Bulk density | High, medium, low | 1, 0.5, 0 |
| Organic carbon content | Very high, high, medium, low, very low | 1, 0.66, 0.5, 0.33, 0 |
| Gypsum content | Yes, no | -1, 0 |

**Table 3. Categorization of each soil characteristic found in each of the eight types of soil orders found in the territory that comprises Argentina.**

| Soil Order | Characteristics of each order of soil found in Argentina (Acidity; Bulk density; Organic carbon content; Gypsum content) |
|---|---|
| Alfisols | -1; 0; 0; 0 |
| Aridisols | 1; 0; 0; -1 |
| Entisols | 0; 0; 0.5; 0 |
| Inceptisols | 0; 0; -1; 0 |
| Mollisols | 1; 0.5; 0.66; 0 |
| Oxisols | 1; 0; 1; 0 |
| Ultisols | 1; 0.5; 0.33; 0 |
| Vertisols | 0; 0; 0.33; 0 |

## Risk model representation

As a first approach, a Red Green Blue (RGB) color risk map was developed to describe the predominant characteristic that increases the risk of STHs. Therefore, the risk associated to SEC is represented in red, to EC it is green and to SC it is blue. The combination obtained from the mix of these colors for each city represents the overall risk associated to STHs so that the preponderance of each characteristic may be observed since although all of these characteristics are important for transmission, we don´t know the importance of one over the other.

To evaluate the effectiveness and coherence of the risk models obtained, the maps were compared with the risk from 20 cities with known prevalence for STH, based on different prevalence studies conducted in 11 of the 23 provinces that make up the country. The prevalence studies included (Table 4) are those performed in areas without MDA interventions and conducted at the community or school level, i.e. not in hospital settings. Prevalence data were ranked in four categories based on WHO guidelines [36], with a prevalence of ≤10% considered as low, a prevalence between 10% and 20% considered as moderate, a prevalence ≥20% and <50% considered as high and finally a prevalence ≥50% considered as very high.

Then, a machine learning technique, concretely a decision tree (DT) classification model, which tries to discriminate classes of objects according to their characteristics in successive steps, was used to obtain a final STH risk. This technique was applied to discriminate low, middle, high and very high prevalence cities. There are different algorithms that make decisions about which variable contributes most significantly in the final classification [37]. Here, a C50 algorithm was trained with the 20 cities mentioned above, using the three types of characteristics as attributes to discriminate prevalence categories (C5.0 R statistical package) [38,39]. As a measure of model accuracy, the observed error was compared the expected distribution for an ordinal variable.

## Results

### Risk associated to SEC, EC and SC

Twenty-two experts on STH participated in the survey developed for the AHP in order to determine their importance in relation to the presence of STH infection. The importance of the contribution of each characteristic within each category was obtained with respect to their relationship with STH infection and the remaining variables. That is the relative frequency of response for each value. The characteristics are ordered from least to most important. For SEC, those turned out to be percent population with computer, unemployment rate, percent urban population, percent population with overcrowding, percent population with access to safe drinking water, and percent population with sanitation at home. For EC they were NDVI,

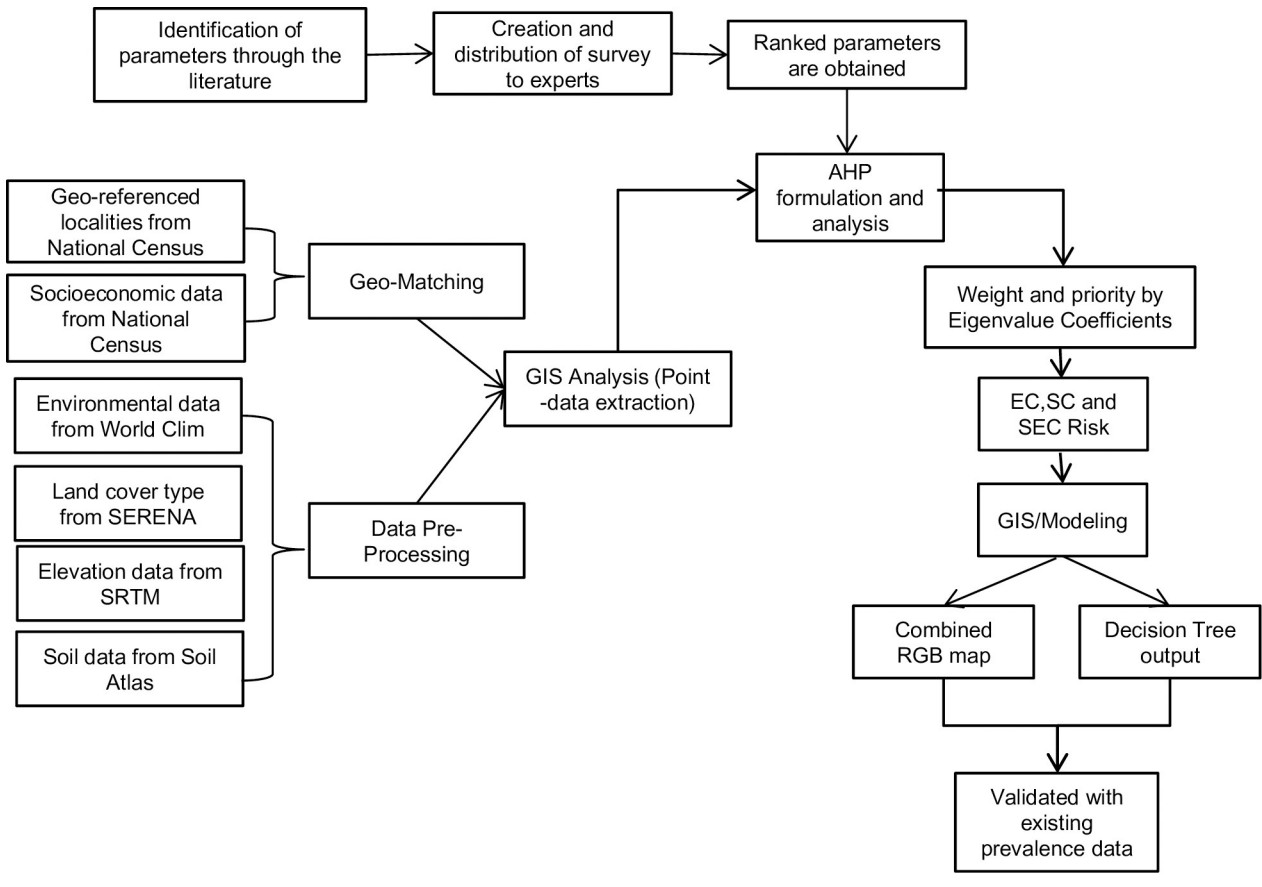

**Fig 1. Summary of the methodology used to obtain and validate the predicted map.**

altitude, land cover, precipitation driest month, precipitation wettest month, maximum temperature, annual precipitation, minimum temperature and mean annual temperature. And finally for SC, they were gypsum content, soil acidity, bulk density and organic carbon content.

From the order obtained, a decision matrix was elaborated, from which the relative importance of each variable was analyzed and therefore, a matrix of eigenvalues was obtained with its corresponding coefficients. Fig 2 shows the weight and the final coefficients for indicators of SEC, EC and SC, reflecting their order of importance with respect to their association with the presence of STH infection.

## Map of risk areas

Separate risk maps were developed for each group of characteristics individually using the coefficients calculated. The risk map for socioeconomic characteristics (Fig 3) shows that the socioeconomic risk is present in many localities of the country, regardless of geographical location or province.

The environmental characteristics map (Fig 4) shows highest risk in the Northeast Argentine region with a gradient of risk that goes from North to South and the lowest risk observed in the Patagonia Region. The Northeast of Argentina is characterized by a subtropical climate, with abundant vegetation (jungle in Misiones province, tall forests in Chaco and Formosa province, low bushes forests in south Corrientes and north Entre Rios province) and large rivers. Therefore, the map is compatible with the land cover areas and characteristics of the country.

**Table 4. Risk values for socioeconomic (SEC), environmental (EC) and soil (SC) characteristics for each city in Argentina with prevalence data for soil-transmitted helminths, observed prevalence categories and predicted categories of risk using the decision tree (DT) algorithm.**

| City | Province | EC risk | SC risk | SEC risk | Prevalence | DT Prediction |
|---|---|---|---|---|---|---|
| Puerto Iguazú[41] | Misiones | 0.8 | 0.5 | 0.3 | Very high | Low |
| Aristóbulo del Valle[42] | Misiones | 0.9 | 1.0 | 0.3 | Very high | Very high |
| Tartagal[43] | Salta | 0.9 | 1.0 | 0.2 | Very high | Very high |
| Misión Km 6[44] | Salta | 0.9 | 1.0 | 0.6 | Very high | Very high |
| San Ramón de la Nueva Orán[45] | Salta | 0.9 | 1.0 | 0.2 | Very high | Very high |
| Bahía Blanca[46] | Buenos Aires | 0.5 | 1.0 | 0.2 | Moderate | Moderate |
| Berisso[47] | Buenos Aires | 0.6 | 1.0 | 0.2 | Low | Moderate |
| General Mansilla[48] | Buenos Aires | 0.7 | 0.0 | 0.2 | Low | Low |
| Coronel Brandsen[49] | Buenos Aires | 0.7 | 1.0 | 0.3 | Moderate | Moderate |
| Santa Fe[50] | Santa Fe | 0.7 | 1.0 | 0.3 | Very high | Low |
| Rosario[51] | Santa Fe | 0.7 | 1.0 | 0.3 | Low | Low |
| Famaillá [52] | Tucumán | 0.8 | 1.0 | 0.1 | High | Low |
| Villa Burruyacú[53] | Tucumán | 0.8 | 1.0 | 0.2 | Moderate | Low |
| San Cayetano[54] | Corrientes | 0.8 | 1.0 | 0.4 | High | Low |
| Santa Ana[55] | Corrientes | 0.8 | 0.6 | 0.3 | Moderate | Low |
| Añatuya[40] | Santiago del Estero | 0.8 | 1.0 | 0.3 | Low | Low |
| Pampa del Indio[56] | Chaco | 0.9 | 0.9 | 0.3 | Low | Moderate |
| Neuquén[57] | Neuquén | 0.5 | 0.9 | 0.1 | Low | Low |
| Córdoba[58] | Córdoba | 0.6 | 0.9 | 0.3 | Moderate | Low |
| Mendoza[59] | Mendoza | 0.7 | 0.9 | 0.0 | Low | Low |

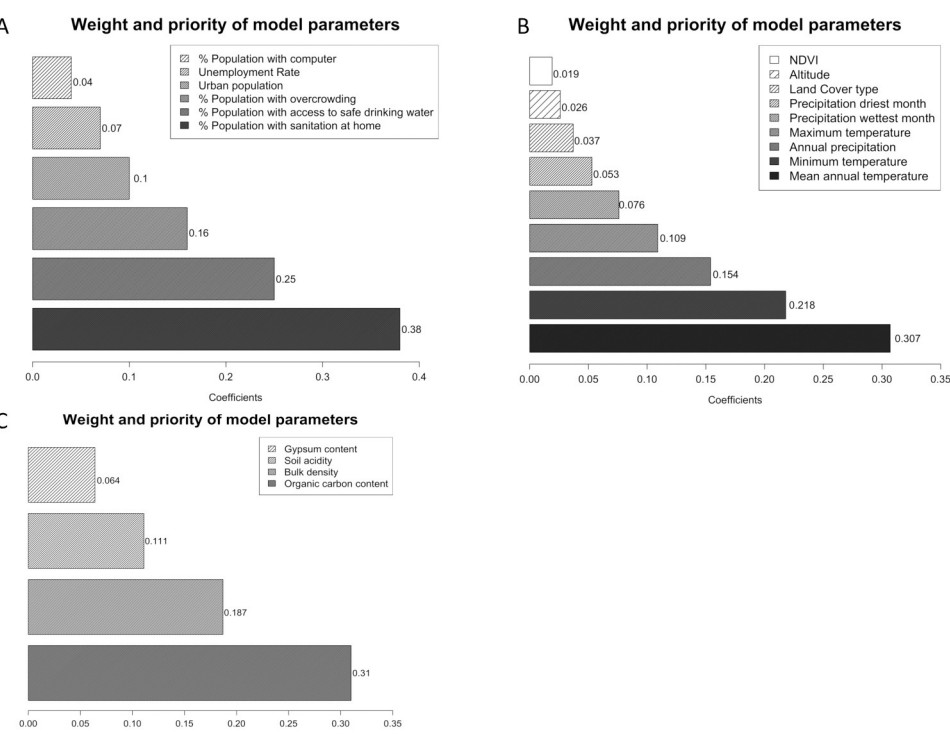

**Fig 2.** Analytical Hierarchy Process coefficients for each indicator of socioeconomic (A), environmental (B) and soil characteristics (C). These values were obtained through the use of a decision matrix from which the relative importance of each variable was analyzed.

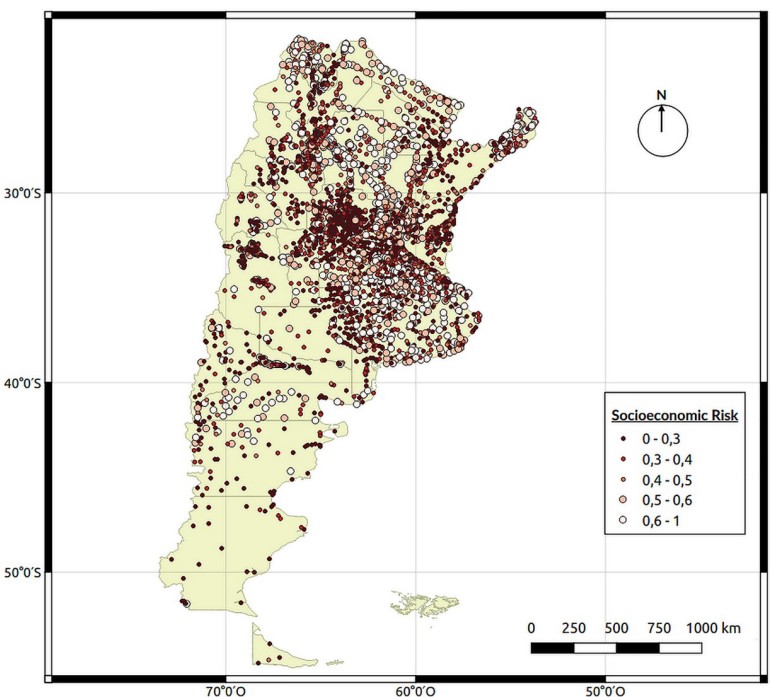

**Fig 3. Socioeconomic (SEC) risk map.** Each point on the map represents a georeferenced city with the average socioeconomic data obtained from the last national census performed in 2010. Map created using QGIS Geographic Information System. Open Source Geospatial Foundation Project. http://qgis.osgeo.org.

Fig 5 shows the risk map for soil characteristics and although the risk is concentrated on the northern half of Argentina, there are some localities with high risk in the southern provinces of Neuquén, Rio Negro and Chubut. This is because of the four soil characteristics included; the one of most relative importance is the content of organic carbon (0.31). The areas shown as high risk are those areas found alongside rivers that cross the Patagonian region, where the soil is much more fertile.

The final RGB map (Fig 6), which takes into account the risk from the individual socioeconomic, environmental and soil characteristics, shows which of the characteristics are more predominant in each locality based on a color scale. Since all of these characteristics are important for the transmission of STHs and all of them need to be present we can observe that the southern area of the country does not have the environmental characteristics adequate for the development of these parasites (reflecting red-violet tones). In the central area of the country, there are some localities between the provinces of Buenos Aires, la Pampa, Córdoba, San Luis and Santa Fe were all three characteristics are present (reflecting pale pink tones). The northern area of the country shows a predominance of appropriate environmental characteristics (reflecting greener tones) with several localities with the presence of all three of them. Fig 7 shows a closer look from two Northern provinces (Formosa and Chaco) to show the examples of localities with a combination of all three characteristics.

## Modeling

The cities analyzed and their prevalence are shown in Table 4, as well as the values of the three risk characteristics modeled for each city, and the prevalence obtained and the predicted results using the DT methodology. The DT obtained by the C50 algorithm identified EC to be

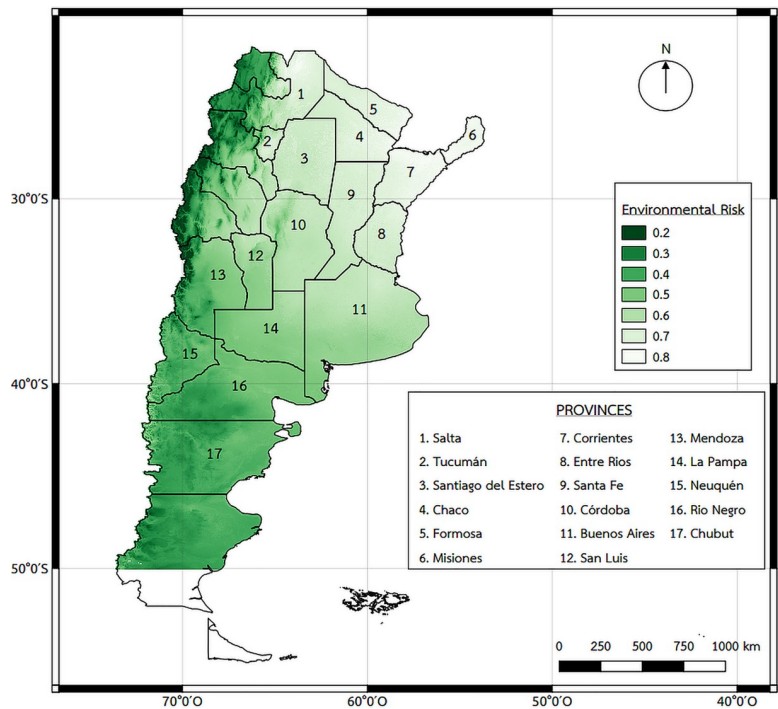

**Fig 4. Environmental characteristics (EC) risk map.** This data was obtained using different sources such as WorldClim which is available until -50˚S latitude, MODIS TERRA NDVI products obtained for a ten-year period (2004 to 2014), the mean of the warmest month calculated through a general vegetation approach (January NDVI mean), the SERENA project for land cover type data and altitude was obtained from the Space Shuttle Radar Topography Mission. Map created using QGIS Geographic Information System. Open Source Geospatial Foundation Project. http://qgis.osgeo.org.

the most important characteristic to classify the 20 cities into the four ordinal categories of risk (expressed as prevalence, Fig 8). The percent of each characteristic used to obtain the final decision tree were as follows: 100% for environmental, 80% for soil and 50% for socioeconomic. The predicted results are shown in Table 4 as "DT Prediction". Using the DT methodology, the expected random error of the ordinal data (four categories) were significantly higher compared with the DT algorithm error (1.25 versus 0.75, p = 0.0004).

Through the use of this algorithm, a total of 247 localities from 5 northern provinces were classified as very high risk: 114 localities from Misiones, 54 localities from Formosa, 34 localities from Salta, 23 localities from Chaco and 22 localities from Corrientes. These are all small localities with an average number of 1,143 inhabitants. The total population at very high risk of infection by STHs would be approximately 1,464,235.

The prevalence obtained in each study through the analysis of fecal samples usually matches the one predicted or varies by one order (i.e. moderate instead of low or vice versa). Nonetheless, in four of the cities included in this study the variation between real and predicted was greater. These four studies, from the cities of Puerto Iguazu, Santa Fe, Famailla and San Cayetano all have high risk with respect to the environment and low risk with respect to the socioeconomic conditions since the INDEC takes the average of each of the cities. The discrepancy here then could be explained by the real SEC of the populations included in the study, which is really much lower than the one reflected by the index since these studies were conducted in marginal areas of the city with no access to running water or improved sanitation.

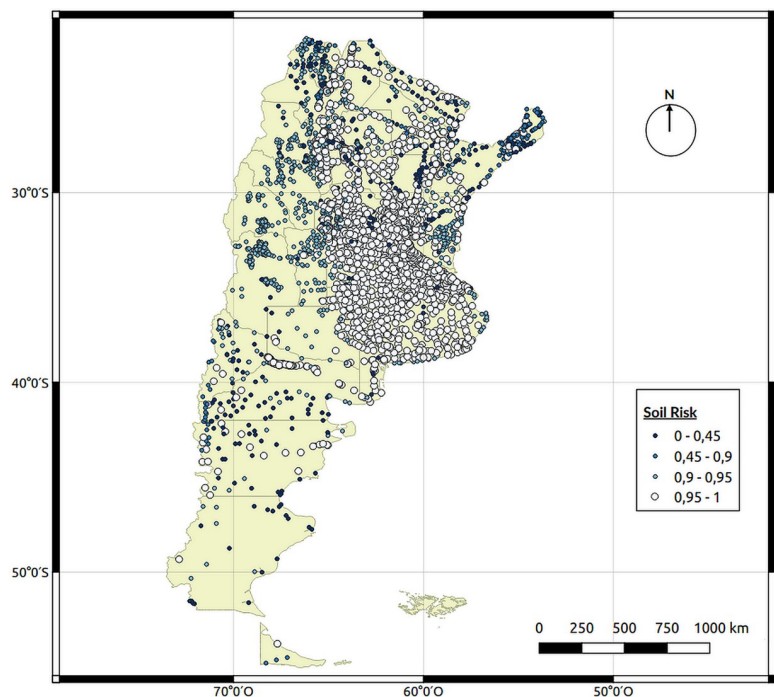

**Fig 5. Soil characteristics (SC) risk map.** Each point represents a city and the different characteristics of the soil were obtained from a Soil Atlas of Argentina from the National Institute of Agricultural Technology. Map created using QGIS Geographic Information System. Open Source Geospatial Foundation Project. http://qgis.osgeo.org.

## Discussion

The lack of prevalence baseline data for STHs in Argentina makes it difficult to determine the need for a control program. A few point prevalence studies conducted in localities from 11 of the 23 provinces that make up the country have shown the presence of all five species of STHs with varying prevalences, ranging from 0.8% to 88.6% [60]. Given the high prevalence found in localities from the Northwest and Northeast of the country and the consequences these infections have on children; it is important to determine the extent of the problem in order to be able to design a tailor-made control program that is implemented either at the national or regional scale. Therefore, the aim of this study was to identify risk areas for the entire country using variables previously associated with the presence of STHs in several studies from other countries, including neighboring Bolivia [13] and Brazil [25].

This is the first risk map for STHs generated specifically for Argentina; it is also one of the few maps generated for this group of parasites that is done at the country scale using socioeconomic, environmental and soil characteristics. The limitation is usually in the availability of census data with socioeconomic characteristics and fortunately for Argentina these were readily available from the last national census. The Analytical Hierarchy Process (AHP) is a common approach used in decision making of complex systems [61–64]. In this study, it has allowed identification of hierarchical relations between separate components (variables identified by a literature review) of a complex problem such as STH transmission. The weight (rank) of the variables within the categories included (socioeconomic, environmental and soil characteristics) were defined by 22 national and international experts in the field of STHs. This result based on expert knowledge may be applied to any geographic region where the SEC, EC and

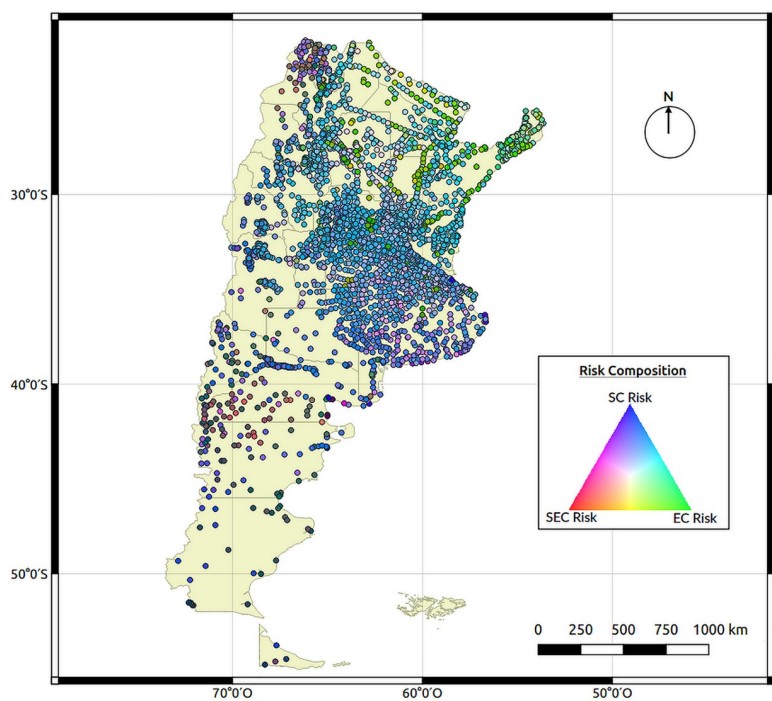

**Fig 6. Final RGB map developed to describe the predominant characteristic that increases the risk of soil transmitted helminths, the risk associated to socioeconomic characteristics is red, to environmental characteristics is green and to soil characteristics is blue.** Each point represents the combination obtained from the mix of these colors for each city. Map created using QGIS Geographic Information System. Open Source Geospatial Foundation Project. http://qgis.osgeo.org.

SC characteristics are available, attempting to standardize a wide range of risk factors used in several previous studies.

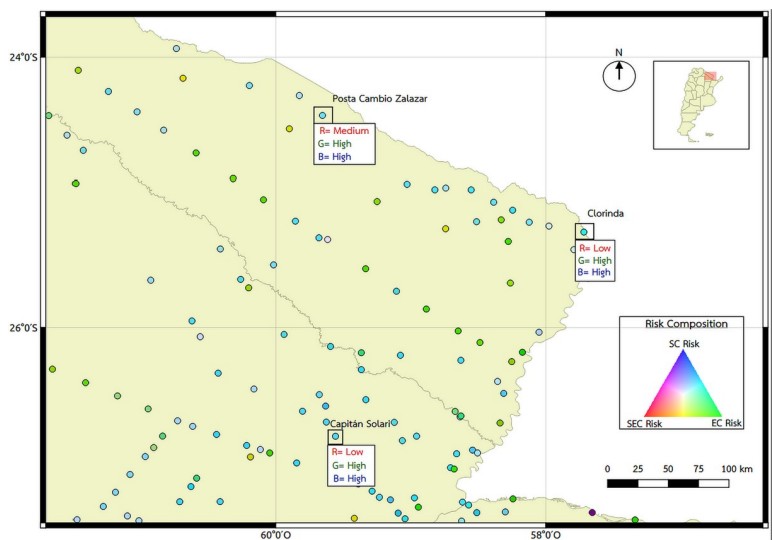

**Fig 7. Description of the values of SEC, EC and SC risk of Northern provinces, two localities from Formosa (Posta Cambio Zalazar and Clorinda) and one from Chaco (Capitán Solari).** Map created using QGIS Geographic Information System. Open Source Geospatial Foundation Project. http://qgis.osgeo.org.

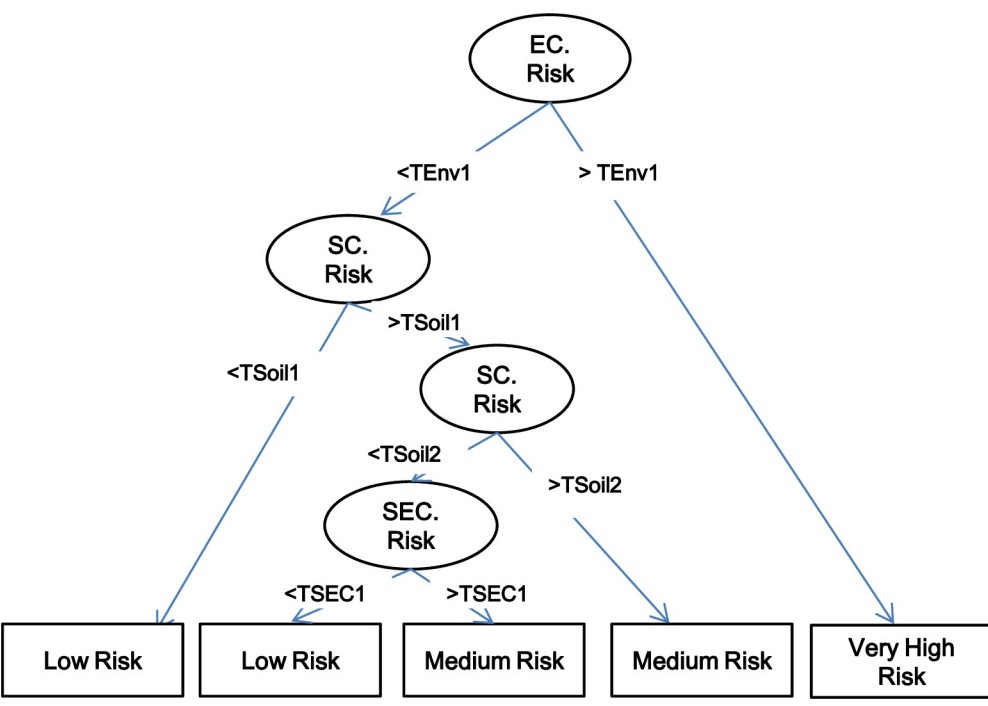

**Fig 8. Hierarchical Decision Tree algorithm obtained with the prevalence training data set from the 20 studies published (Table 4).** Thresholds (T) with respect to the prevalence categories of risk identified in each node (characteristic) which corresponds to the characteristics included in the study, environment (TEnv), soil (TSoil) and socioeconomic (TSec). Squares are the final classification. Given this is a machine learning technique, the values would change responding to the different training sets offered (observed prevalence data).

In Argentina, the resulting individual maps show those areas of the country with the appropriate characteristics for the transmission of STHs. Given certain biological characteristics of these parasites: 1) no animal reservoir, transmission from human to human; 2) life cycle with the development of infective stages in the environment; 3) transmission by the fecal-oral route (*A. lumbricoides* and *T. trichiura*) and by larval penetration (hookworms and *S. stercoralis*), it is important to consider these three categories of characteristics in a combined map in order to more accurately assess their presence. In order to obtain a final STH risk assigning weight or coefficients to the three different characteristics used (EC, SC and SEC) in an objective manner, a Decision Tree (DT) algorithm was used. These DT classifiers are included into machine learning techniques were the number and quality of training data sets are crucial. Therefore, increasing the number of sites (cities) will produce more accurate predictions of final STH risk. Nonetheless, the error measure obtained is about 50% lower than those expected by random assignment of the four risk prevalence categories (low, moderate, high, very high)

The combined map generated was validated using prevalence data from the different point studies conducted and in general it was able to correctly predict the prevalence of STHs in 15 of the 20 studies included. This evidences certain limitation with have been previously addressed in other studies, such as the parasitology method used for diagnosis [26], variations of socioeconomic conditions within localities and the unavailability to measure certain soil characteristics using remote sensing (i.e. salt content) [13]. Moreover, uncertainty measurements need to be considered. In the case of the three components used in this study, we can assume that they would be the sum of the error of each individual one. Therefore, the percent

relative error for the social [29,65] and environmental characteristics is around 10% [30,31,66]. On the other hand, the error for the soil is larger since the scale available in the maps is an approximation based on soil order and there is a lack of information on the precise composition of the soil with respect to the variables of interest (gypsum content, soil acidity, bulk density and organic carbon content) [34].

The resulting RGB map was generated so that the contribution of each category of characteristics used can be inferred, some areas have only two of the three categories, i.e. SEC and EC, or EC and SC, and those areas that show a combination of all three would be the ones to prioritize for the conduction of baseline surveys to determine the presence of STHs. According to this map, the top half of the country, from Central Argentina to the North, has localities with the characteristics necessary for the development of these parasites. Using the DT methodology, a total of 1,464,235 people would be at very high risk of infection by STHs. With the current availability of geospatial and environmental data and the tendency of an open data policy, the work scheme presented herein for the development of an STH risk map, is highly reproducible in other areas.

## Conclusion

The predictive map generated takes into consideration the combination between socioeconomic, environmental and soil characteristics that are more closely associated with the risk of soil-transmitted helminths as classified by the use of a DT methodology and therefore should serve as a useful tool for guiding the identification of survey areas for the generation of baseline data, detecting hotspots of infection, planning and prioritizing areas for control interventions, and eventually performing post-implementation surveillance activities.

## Acknowledgments

We would like to thank Marcelo Scavuzzo and Sofía Lanfri from the Mario Gulich Institute for Higher Space Studies, Teófilo Tabanera Space Center, Córdoba, Argentina for their valuable guidance/input in the early stages of this study. We would also to thank the 22 experts from the STH community, including STH Coalition members, who took the time to answer the survey for this study.

## Author Contributions

**Conceptualization:** Marcelo C. Abril, Ximena Porcasi, María V. Periago.

**Data curation:** Eliana M. Alvarez Di Fino, Ximena Porcasi, María V. Periago.

**Formal analysis:** Eliana M. Alvarez Di Fino, María V. Periago.

**Funding acquisition:** Marcelo C. Abril.

**Methodology:** Ximena Porcasi, María V. Periago.

**Project administration:** María V. Periago.

**Resources:** Jorge Rubio.

**Software:** Jorge Rubio.

**Supervision:** Marcelo C. Abril.

**Writing – original draft:** Eliana M. Alvarez Di Fino, Jorge Rubio, María V. Periago.

**Writing – review & editing:** Eliana M. Alvarez Di Fino, Marcelo C. Abril, Ximena Porcasi, María V. Periago.

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
