## [Decision Letter · Decision Letter 0]

19 Sep 2019

Dear Dr. Periago:

Thank you very much for submitting your manuscript "Risk map development for soil-transmitted helminth infections in Argentina" (#PNTD-D-19-01254) for review by PLOS Neglected Tropical Diseases. Your manuscript was fully evaluated at the editorial level and by independent peer reviewers. The reviewers appreciated the attention to an important problem, but raised some substantial concerns about the manuscript as it currently stands. These issues must be addressed before we would be willing to consider a revised version of your study. We cannot, of course, promise publication at that time.

We therefore ask you to modify the manuscript according to the review recommendations before we can consider your manuscript for acceptance. Your revisions should address the specific points made by each reviewer. 

When you are ready to resubmit, please be prepared to upload the following:

(1) A letter containing a detailed list of your responses to the review comments and a description of the changes you have made in the manuscript.

(2) Two versions of the manuscript: one with either highlights or tracked changes denoting where the text has been changed (uploaded as a "Revised Article with Changes Highlighted" file); the other a clean version (uploaded as the article file).

(3) If available, a striking still image (a new image if one is available or an existing one from within your manuscript). If your manuscript is accepted for publication, this image may be featured on our website. Images should ideally be high resolution, eye-catching, single panel images; where one is available, please use 'add file' at the time of resubmission and select 'striking image' as the file type. 

Please provide a short caption, including credits, uploaded as a separate "Other" file. If your image is from someone other than yourself, please ensure that the artist has read and agreed to the terms and conditions of the Creative Commons Attribution License at http://journals.plos.org/plosntds/s/content-license (NOTE: we cannot publish copyrighted images). 

(4) If applicable, we encourage you to add a list of accession numbers/ID numbers for genes and proteins mentioned in the text (these should be listed as a paragraph at the end of the manuscript). You can supply accession numbers for any database, so long as the database is publicly accessible and stable. Examples include LocusLink and SwissProt.

(5) To enhance the reproducibility of your results, we recommend that you deposit your laboratory protocols in protocols.io, where a protocol can be assigned its own identifier (DOI) such that it can be cited independently in the future. For instructions see http://journals.plos.org/plosntds/s/submission-guidelines#loc-methods

While revising your submission, please upload your figure files to the Preflight Analysis and Conversion Engine (PACE) digital diagnostic tool, https://pacev2.apexcovantage.com/ PACE helps ensure that figures meet PLOS requirements. To use PACE, you must first register as a user. Then, login and navigate to the UPLOAD tab, where you will find detailed instructions on how to use the tool. If you encounter any issues or have any questions when using PACE, please email us at figures@plos.org.

We hope to receive your revised manuscript by Nov 18 2019 11:59PM. If you anticipate any delay in its return, we ask that you let us know the expected resubmission date by replying to this email.

To submit a revision, go to https://www.editorialmanager.com/pntd/ and log in as an Author. You will see a menu item call Submission Needing Revision. You will find your submission record there. 

Sincerely,

Martin Walker

Guest Editor

Jennifer Keiser

Deputy Editor

Reviewer's Responses to Questions

**Key Review Criteria Required for Acceptance?**

**Methods**

-Are the objectives of the study clearly articulated with a clear testable hypothesis stated?

-Is the study design appropriate to address the stated objectives?

-Is the population clearly described and appropriate for the hypothesis being tested?

-Is the sample size sufficient to ensure adequate power to address the hypothesis being tested?

-Were correct statistical analysis used to support conclusions?

-Are there concerns about ethical or regulatory requirements being met?

Reviewer #1: The method used in not completely new but very clearly explained . The verification of the results of the model with real data is an important aspect of the paper.

Reviewer #2: Limitations of the methods have not been addressed.

**Results**

-Does the analysis presented match the analysis plan?

-Are the results clearly and completely presented?

-Are the figures (Tables, Images) of sufficient quality for clarity?

Reviewer #1: In the results session (Table 4) the authors are comparing the 3 indexes with the real results; in my opinion this is not ideal because is difficult to understand when the model predicted well and where failed to predict.

For example San Caytano SEC=0.44; EC=0.7 SC=1 ; is not clear to me what is the prediction from the model in term of STH prevalence? One index (SEC) is the lower of the list in the list one other is the higher (SC) 

I think that the authors, in addition to “SEC Risk” “EC Risk” and “SC risk”, should also provide a composite index that predicts a unique STH prevalence.

I do not think the thee indexes are equally important in predicting the STH prevalence and therefore the 3 indexes can be given different weight in the formula.

Then in table 4 I would provide for each city a “predicted STH prevalence” (low - moderate - high - very high) and a “real STH prevalence” (low - moderate - high - very high) This will make very simple to evaluate if the prediction modeled were verified in reality.

The authors attempt do what I suggested in figure 9 but in my opinion this is not clear which color correspond to high risk? There is no legend clarifying this the color-code triangle in my opinion is not helpful because it give only the proportion of the different indexes. Which color will be a city A in which all the 3 indexes are predicting consistently high prevalence? And which color in a city B in which all the indexes predict consistently low prevalence? Probably with the method used by the author the two cities will have similar color…

Reviewer #2: The study lacks uncertainty measures.

**Conclusions**

-Are the conclusions supported by the data presented?

-Are the limitations of analysis clearly described?

-Do the authors discuss how these data can be helpful to advance our understanding of the topic under study?

-Is public health relevance addressed?

Reviewer #1: coclusion are OK, but the method should be improved as suggested

Reviewer #2: Validation methods should be discussed further and limitations should be disclosed.

**Editorial and Data Presentation Modifications?**

Reviewer #1: Minor comments:

• Page 8 I would use “Preventive chemotherapy” instead of “Mass Drug Administration” because the Mass Drug Administration is a form of Preventive chemotherapy that is aiming to cover the entire population, while in the case of STH only groups at risk are recommended for treatment.

• After the first citation please refers to the STH species as A. lumbricoides or T. trichiura and not as Ascaris or Trichuris

Reviewer #2: (No Response)

**Summary and General Comments**

Reviewer #1: The article address a relevant issue in the field of NTD, is well written and in my opinion merits publication with minor revisions

Reviewer #2: Authors are presenting a risk map of soil-transmitted helminths (STH) in Argentina developed using an analytical hierarchy process were 20 experts in the field decided the importance of socioeconomic, environmental (climates and altitude), and soil variables on the development of STHs infections. Although the study is addressing a gap in knowledge, they should improve their discussion and present uncertainty measures. Some recommendations based on sections since the paper lacked page numbers or lines: 

Introduction

Paragraph 1 and 2 should be together, they are talking about the same thing. If you want to split them, consider separating them in ‘For control…’

Paragraph 2: In the last sentence ‘it is not included’ affirmation is referring to WHO guidelines for deworming? 

Last paragraph: Refer to table 2 where all the references of STHs studies in Argentina are presented. 

Methods: 

Since it is the only equation, I would suggest eliminating the number inside square brackets (i.e., [1]). 

Data collection: 

Worldclim does not offer soil variables, only climatic data for temperature and precipitation. Which pixel size (resolution) of worldclim variables was used? This is key since you will be having different values in your analysis if you are using 5-minute arc, 2.5-minute arc, etc. 

There is not enough detail of the type of NDVI data collected. You should describe the resolution of the variables, the source of the images, and also the satellite from where the index was derived; it is different to use 500-meter biweekly resolution variables than 1 km monthly variables considering that you are using averages. Also, it is important to detail the source (NASA-Earth data?), the version (e.g., 006), and the satellite from where they were derived (Terra, Aqua, the products from both satellites?). 

Please describe SERENA land cover variables briefly, are those a categorical product showing different classes of environments at percentages? What is the resolution of the altitude variable from SRTM? 

Please provide a brief description for the diversity index developed for the cities studied.

Table 1: 

Does the variable urban population should be % of urban population as in the main text?

Figures: 

It is recommended that all the figures follow equal features, currently figure 1 and three have not frames but figure 2 and 4 have them. North arrow is different depending on the figures. 

Also, consider adding Argentinean provinces in all maps (currently only in figures 8-9) and potentially label the provinces discussed in the main text. 

Why Fig 6 (Environments) is presented as a continuous map while Figs 5 and 7 are presented as points? This question is relevant because the RGB map (Figs 8 and 9) is showing environments as point values again. Is Fig 6 interpolating values? 

Add province names in Figs 9-11. 

Figures 1-3 can be combined in one figure 

Figures 4-6 can be combined in one figure 

Figures 10-11 can be combined in one figure 

Results and discussion: 

Would you be able to present uncertainty measures for your predictions? I am mentioning this because your model validated 15 over 20 localities where prevalence data is available, however, you applied your model to 3526 cities. Thus, it will be useful to address how good is your prediction by considering points without available information. 

Consider also discussing limitations of your approach, at present you are not declaring any potential problem with your AHP including variable selection and expert opinion. Further, prevalence of the papers considered in this study for validation might also be biased considering their own limitations and diagnostic methods; this should be also explicit to assure a fair treatment of their analysis. 

Others: 

Consider that the species status of Ascaris lumbricoids and A. suum is being currently discussed posing a potential zoonotic feature of the genus Ascaris spp. (Review: https://doi.org/10.1017/S0022149X18000160 and https://doi.org/10.1186/1756-3305-5-42)

Review references, for example, reference 3 starts with a year. Some references are mentioning the name of the article twice.

PLOS authors have the option to publish the peer review history of their article (what does this mean?). If published, this will include your full peer review and any attached files.

Reviewer #1: Yes: Antonio Montresor

Reviewer #2: No

---

## [Editor Report · Decision Letter 1]

27 Nov 2019

Dear Dr. Periago:

Thank you very much for submitting your manuscript "Risk map development for soil-transmitted helminth infections in Argentina" (PNTD-D-19-01254R1) for review by PLOS Neglected Tropical Diseases. Your manuscript was fully evaluated at the editorial level and by independent peer reviewers. The reviewers appreciated the attention to an important topic but identified some aspects of the manuscript that should be improved.

We therefore ask you to modify the manuscript according to the review recommendations before we can consider your manuscript for acceptance. Your revisions should address the specific points made by each reviewer.

(1) A letter containing a detailed list of your responses to the review comments and a description of the changes you have made in the manuscript.

(2) Two versions of the manuscript: one with either highlights or tracked changes denoting where the text has been changed (uploaded as a "Revised Article with Changes Highlighted" file ); the other a clean version (uploaded as the article file).

(3) If available, a striking still image (a new image if one is available or an existing one from within your manuscript). If your manuscript is accepted for publication, this image may be featured on our website. Images should ideally be high resolution, eye-catching, single panel images; where one is available, please use 'add file' at the time of resubmission and select 'striking image' as the file type. 

Please provide a short caption, including credits, uploaded as a separate "Other" file. If your image is from someone other than yourself, please ensure that the artist has read and agreed to the terms and conditions of the Creative Commons Attribution License at http://journals.plos.org/plosntds/s/content-license (NOTE: we cannot publish copyrighted images). 

(4) Appropriate Figure Files 

Please remove all name and figure # text from your figure files upon submitting your revision. Please also take this time to check that your figures are of high resolution, which will improve both the editorial review process and help expedite your manuscript's publication should it be accepted. Please note that figures must have been originally created at 300dpi or higher. Do not manually increase the resolution of your files. For instructions on how to properly obtain high quality images, please review our Figure Guidelines, with examples at: http://journals.plos.org/plosntds/s/figures

While revising your submission, please upload your figure files to the Preflight Analysis and Conversion Engine (PACE) digital diagnostic tool, https://pacev2.apexcovantage.com/ PACE helps ensure that figures meet PLOS requirements. To use PACE, you must first register as a user. Then, login and navigate to the UPLOAD tab, where you will find detailed instructions on how to use the tool. If you encounter any issues or have any questions when using PACE, please email us at figures@plos.org.

We hope to receive your revised manuscript by Jan 26 2020 11:59PM. If you anticipate any delay in its return, we ask that you let us know the expected resubmission date by replying to this email.

To submit your revised files, please log in to https://www.editorialmanager.com/pntd/

Sincerely,

Martin Walker

Guest Editor

Jennifer Keiser

Deputy Editor

Line 105: Eigenvalue matrix – what does this mean? Do the Authors mean the eigenvalue of a matrix? I suggest using a more accessible description. This description also appears on line 190. It will not be clear to the readership what this means. 

Line 107: change lineal to linear

Line 114, 122 and elsewhere: “data” is a plural noun, therefore the text should read “data are”

Line 125: Could the authors clarify what “NDVI products” means?

Line 128: I believe the word “approach” should read “index”. 

Lines 125-139: In the interests of clarity it would be helpful if the Authors could give the resolution of the different forms of remotely sensed data in standard and consistent units. 

Line 154: delete “Therefore”

Line 173: Could the Authors please clarify or give further detail on how model accuracy was quantified? The current wording, “Measure of model accuracy was obtained comparing the expected statistic of a variable distribution of the error with the four categories of

ordinal data and the observed error value with its p-value.” is not clear.

Lines 182- 184: Please use “percentage” rather than “%”

Lines 185-188: Please review the use of capital letters. 

The legends of Figures 2 to 7 are too brief and mainly consist of a title only. Please remember that the figure legend is supposed to give (brief) relevant information to allow a reader to understand the main message in isolation from the text. For example, in Figures 5, 6 and 7, the meaning of the points should be described. Are these populations? 

Figure 5. It is not entirely clear to me why soil-risk was not presented continuously (as environmental risk in Figure 4). Is this an issue with the availability of data or does it relate to where human populations are? 

Line 240: I do not think the machine learning approach describe constitutes a “model validation”. It seems that the Authors have combine the environmental/socioeconomic risks with prevalence data to fit a decision tree model. This is fine but it is not validation. That would be testing the model predictions against prevalence data that were not used for training/fitting. 

Line 286: Can the Authors specify which STH species have been identified in Argentina?

Line 320: Again, I do not think the predictions have been “validated” – as the Authors state, the prevalence data used is key to informing the predictive power of the model but, as I understand, no data were withheld from training to be used for validation. 

Since the Authors have recent census data for Argentina, the manuscript would be strengthened by enumerating the population living in low, moderate and high risk (high prevalence) areas.

---

## [Editor Report · Decision Letter 2]

17 Dec 2019

Dear Dr. Periago,

We are pleased to inform you that your manuscript, "Risk map development for soil-transmitted helminth infections in Argentina", has been editorially accepted for publication at PLOS Neglected Tropical Diseases.

Before your manuscript can be formally accepted and sent to production you will need to complete our formatting changes, which you will receive in a follow up email. Please note: your manuscript will not be scheduled for publication until you have made the required changes.

IMPORTANT NOTES

* Copyediting and Author Proofs: To ensure prompt publication, your manuscript will NOT be subject to detailed copyediting and you will NOT receive a typeset proof for review. The corresponding author will have one final opportunity to correct any errors when sent the requests mentioned above. Please review this version of your manuscript for any errors.

* If you or your institution will be preparing press materials for this manuscript, please inform our press team in advance at plosntds@plos.org. If you need to know your paper's publication date for media purposes, you must coordinate with our press team, and your manuscript will remain under a strict press embargo until the publication date and time. PLOS NTDs may choose to issue a press release for your article. If there is anything that the journal should know, please get in touch.

*Now that your manuscript has been provisionally accepted, please log into EM and update your profile. Go to http://www.editorialmanager.com/pntd, log in, and click on the "Update My Information" link at the top of the page. Please update your user information to ensure an efficient production and billing process.

*Note to LaTeX users only - Our staff will ask you to upload a TEX file in addition to the PDF before the paper can be sent to typesetting, so please carefully review our Latex Guidelines [http://www.plosntds.org/static/latexGuidelines.action] in the meantime.

Best regards,

Martin Walker

Guest Editor

Jennifer Keiser

Deputy Editor

---

## [Editor Report · Acceptance letter]

23 Jan 2020

Dear Dr. Periago,

We are delighted to inform you that your manuscript, "Risk map development for soil-transmitted helminth infections in Argentina," has been formally accepted for publication in PLOS Neglected Tropical Diseases.

Best regards,

Serap Aksoy

Editor-in-Chief

Shaden Kamhawi

Editor-in-Chief
